# Interleukin-6 Induces Stem Cell Propagation through Liaison with the Sortilin–Progranulin Axis in Breast Cancer

**DOI:** 10.3390/cancers15245757

**Published:** 2023-12-08

**Authors:** Karoline Berger, Emma Persson, Pernilla Gregersson, Santiago Ruiz-Martínez, Emma Jonasson, Anders Ståhlberg, Sara Rhost, Göran Landberg

**Affiliations:** 1Department of Laboratory Medicine, Institute of Biomedicine, Sahlgrenska Academy, University of Gothenburg, 40530 Gothenburg, Sweden; karoline.berger0809@gmail.com (K.B.); emmahpersson@gmail.com (E.P.); pernilla.gregersson@yahoo.se (P.G.); santiago.ruiz.martinez@gu.se (S.R.-M.); emma.jonasson@llcr.med.gu.se (E.J.); anders.stahlberg@gu.se (A.S.); sara.rhost@gu.se (S.R.); 2Wallenberg Center for Molecular and Translational Medicine, University of Gothenburg, 41390 Gothenburg, Sweden; 3Department of Clinical Genetics and Genomics, Sahlgrenska University Hospital, 41346 Gothenburg, Sweden

**Keywords:** breast cancer, cancer stem cells, progranulin, sortilin, interleukin-6, interleukin-8

## Abstract

**Simple Summary:**

Accumulating studies indicate that cancer stem cells are responsible for drug resistance, metastasis and tumor recurrence of aggressive breast cancers due to intrinsic aggressive features. The aim of our study was to identify secreted molecules important for the breast cancer stem cell population and to determine a possible link with the previously described cancer stem cell activator progranulin and its receptor, sortilin. In monolayer cultures, progranulin induced secretion of several inflammatory-related cytokines, such as interleukin (IL)-6 and -8, in a sortilin-dependent manner, revealing a distinct interplay between progranulin, IL-8 and IL-6. Using models for cancer stem cell features and patient-based cancer microenvironments, we could validate the effect of IL-6 and progranulin in promoting cancer stem cells via the sortilin receptor. The results highlight sortilin as a highly relevant therapeutic target for aggressive breast cancer.

**Abstract:**

Unraveling the complex network between cancer cells and their tumor microenvironment is of clinical importance, as it might allow for the identification of new targets for cancer treatment. Cytokines and growth factors secreted by various cell types present in the tumor microenvironment have the potential to affect the challenging subpopulation of cancer stem cells showing treatment-resistant properties as well as aggressive features. By using various model systems, we investigated how the breast cancer stem cell-initiating growth factor progranulin influenced the secretion of cancer-associated proteins. In monolayer cultures, progranulin induced secretion of several inflammatory-related cytokines, such as interleukin (IL)-6 and -8, in a sortilin-dependent manner. Further, IL-6 increased the cancer stem fraction similarly to progranulin in the breast cancer cell lines MCF7 and MDA-MB-231 monitored by the surrogate mammosphere-forming assay. In a cohort of 63 patient-derived scaffold cultures cultured with breast cancer cells, we observed significant correlations between IL-6 and progranulin secretion, clearly validating the association between IL-6 and progranulin also in human-based microenvironments. In conclusion, the interplay between progranulin and IL-6 highlights a dual breast cancer stem cell-promoting function via sortilin, further supporting sortilin as a highly relevant therapeutic target for aggressive breast cancer.

## 1. Introduction

Cancer stem cells are a small subpopulation of undifferentiated cancer cells with clear malignant properties such as a capacity for self-renewal and an inherent resistance to cancer therapies [1,2]. When studying the complexity of cancer progression, including cancer stem cells as well as inflammatory signaling, it is obvious that the cancer niche and the specific cancer microenvironment influence aggressive features of the disease [3,4]. Cytokines, growth factors and other proteins secreted by various cells in the tumor microenvironment specifically influence key features linked to cancer metastases also associated with breast cancer stem cell properties [5,6,7,8]. However, for proper understanding of the complex communication between cancer stem cells and the tumor microenvironment, it is important to use optimal in vivo-based experimental platforms. Findings based on the entire cancer niche can be used to decipher novel cancer-progressing mechanisms and to identify more efficient treatment strategies, including the targeting of breast cancer stem cells.

The pleiotropic growth factor progranulin is an 88 kDa glycoprotein implicated in numerous biological processes, such as proliferation, wound repair, survival and inflammation [9]. Importantly, progranulin is overexpressed in various types of cancers, including breast cancer [10]. In addition, progranulin has been linked to malignant cell transformation, migration and invasion, as well as tumor growth and therapy resistance [10,11]. We recently demonstrated that breast cancer cells exposed to a hypoxic environment secreted progranulin, and that this led to an increase in the cancer stem cell fraction in vitro, as well as in the metastatic ability of cancer cells in vivo [12]. Known receptors for progranulin are the neuronal receptor sortilin [13], as well as the inflammatory receptors tumor necrosis factor receptor (TNFR)-I and TNFRII [14] and the newly identified receptor Ephrin A2 (EphA2) [15]. Sortilin is involved in protein sorting and trafficking, shuttling between the cell surface and various intracellular compartments, where it translocates proteins to distinct destinations [16]. Further, sortilin mediates internalization through endocytosis [17] and has been associated with cancer development [18], and more specifically with breast cancer aggressiveness [19]. Sortilin has also been suggested to be involved in the direct interaction and production of various cytokines, including interleukin (IL)-6 in immune cells [20,21].

In order to better understand the impact of progranulin-driven cancer stem cell propagation, we analyzed the secretomic profile of breast cancer cells treated with progranulin and identified a crosstalk between the progranulin–sortilin axis and IL-6 in different model systems, including a human-based 3D-like growth platform. A better understanding of the complex network between cancer cells and their tumor environment is of immense importance, as it might result in the identification of novel therapeutic targets and more effective treatment regimens for breast cancer patients.

## 2. Materials and Methods

### 2.1. Cell Lines and Culture Conditions

The MCF7 and MDA-MB-231 breast cancer cell lines were purchased from the American Type Culture Collection (ATCC, Manassas, VA, USA). MCF7 cells were sustained and grown as monolayers in DMEM (Lonza, Basel, Switzerland) containing 10% fetal bovine serum (FBS, Thermo Fisher Scientific, Waltham, MA, USA), 1% non-essential amino acids, 1% L-Glutamine and 1% Penicillin/Streptomycin (Thermo Fisher Scientific), and MDA-MB-231 cells were sustained in RPMI 1640 medium (Lonza) supplemented with 10% FBS, 1% Sodium Pyruvate, 1% L-Glutamine and 1% Penicillin/Streptomycin (Thermo Fisher Scientific) and maintained in a humidified incubator at 37 °C with an atmospheric pressure of 5% (*v*/*v*) CO_2_. All cell lines were validated as mycoplasma-free using a Mycoplasma PCR kit (Applied Biological Materials, Richmond, BC, Canada) in house and authenticated by ATCC using short tandem repeat profiling.

### 2.2. Drug Treatments

Recombinant human progranulin (AG-40A-0068Y, Adipogen, San Diego, CA, USA) was reconstituted in sterile PBS upon delivery, aliquoted and stored at −20 °C. Recombinant human IL-6 and IL-8 (#206-IL and #208-IL, R&D Systems, Minneapolis, MN, USA) were reconstituted, as recommended, in sterile PBS with 0.1% BSA and stored at −20 °C. The AF38469 sortilin inhibitor (MedChem Express, Monmouth Junction, NJ, USA) was reconstituted in DMSO to a stock concentration of 132.6 mM. Working stock concentrations were achieved by dilution in culture media. Cell lines were treated with specified concentrations of the respective compounds and controls for 48 h at 37 °C with 5% CO_2_ and 21% O_2_, unless stated otherwise.

### 2.3. Proximity Extension Assay (PEA)

To target the proteomics of secreted proteins, proximity extension assay (PEA) technology was performed (Olink Proteomics, Uppsala, Sweden). The PEA technology is based on oligonucleotide-labeled antibodies bound to the target protein, leading to hybridization, followed by amplification and real-time PCR. To obtain conditioned media for PEA analysis, cells were treated for 48 h with 1 μg/mL progranulin, 2 μg/mL AF38649 or both. Conditioned media were collected, centrifuged at 300× *g* for 3 min and stored at −80 °C, until the samples were sent for PEA analysis. The Immuno-oncology and Cardiovascular III Olink panels (Olink Proteomics) were used. To compensate for protein in cell media serum, a media control was subtracted and normalized between the different samples (mean-centered).

### 2.4. Western Blot

MCF7 and MDA-MB-231 cells were treated with respective proteins (progranulin, IL-6 or IL-8) for 48 h. Cells were lysed in RIPA lysis buffer (Sigma-Aldrich, St. Louis, MO, USA) and separated on a 4–20% SDS-PAGE gel (Mini-PROTEAN^®^ TGX^TM^ Precast Gels, 10 or 15 wells (Bio-Rad, Hercules, CA, USA), followed by transfer to a nitrocellulose membrane 0.45 μm, (#88018, Thermo Fisher Scientific) or 0.2 μm, (#162-0146, Bio-Rad). SYPRO™ Ruby Protein Blot Stain (S11791, Thermo Fisher Scientific) was used as a loading control. The nitrocellulose membrane was probed with a primary anti-progranulin (0.75 μg/mL, AF2420, R&D Systems), anti-IL-8 (1:1000, ab18672, Abcam, Cambridge, UK), anti-IL-6 (1:1000, ab9324, Abcam) or anti-sortilin antibody (1:1000, ab1660, Abcam), followed by incubation with applicable horseradish peroxidase-conjugated secondary antibodies (1:5000, anti-mouse/rabbit/goat HRP conjugated, R&D Systems). Protein detection was carried out using SuperSignal^TM^ West Femto Maximum Sensitivity Substrate (#34095, Thermo Fisher Scientific), and chemoluminescence was detected using AMERSAM^TM^ ImageQuant^TM^ 800 (GE Healthcare, Chicago, IL, USA). SYPRO™ Ruby staining was detected using UV transillumination on a Gel Doc Imaging System (Version 5.2, Bio-Rad). Images were processed using ImageJ Software (1.52p, W. Rasband, National Institutes of Health), and band density was compared relative to SYPRO™ Ruby staining.

### 2.5. Mammosphere Formation Assay

The mammosphere formation assay was performed as described previously [22]. Briefly, single-cell suspensions were obtained following treatment with respective compounds and seeded in phenol red-free DMEM/F-12 (Gibco^®^, Grand Island, NY, USA), supplemented with 1% B27 (Thermo Fisher Scientific), 1% P/S and 20 ng/mL EGF (BD Biosciences, Franklin Lakes, NJ, USA) onto non-adherent poly-2-hydroxyethyl methacrylate (polyHEMA)-coated plates. After cultivation for five to seven days, spheres greater than 50 μm in diameter were counted manually under the microscope.

### 2.6. Alamar Blue Viability Assay

Cell viability was determined by an Alamar Blue-based metabolic assay according to the manufacturer’s instructions (Thermo Fisher Scientific). Cells were seeded as triplicates in 96-well plates (5000 cells/well) and treated with different compound concentrations for 48 h. A baseline analysis was performed prior to treatment as well as 48 h post-treatment to measure the relative viability of the cells. Alamar Blue reagent was added to each well, and absorbance was analyzed using a VICTOR3 Multilabel Plate Reader (Perkin Elmer Life Sciences, Waltham, MA, USA).

### 2.7. Competative Fluorescent Polarization Assay (FPA)

For analyzing the binding of proteins in a competitive sortilin-binding assay, the extracellular part of human sortilin (NM002959.6/Q99523), amino acids 1–756 plus a C-terminal His6-tag, was produced in CHO-S cells by transient transfection as a secreted protein. Supernatants from two different transfection reagents were pooled: 150 mL FectoPro and 150 mL NovaCHOice. Purification was performed using Immobilized Metal Ion Affinity Chromatography (IMAC) (Mammalian Protein Expression (MPE) core facility, Sahlgrenska Academy) in buffer (50 mM Hepes pH 7.4, 100 mM NaCl and 2 mM CaCl_2_). Proteins were eluted using an imidazole-gradient (125–500 mM). Fractions containing sSORT were pooled, and protein size was confirmed by Western blot. Buffer was exchanged to 50 mM HEPES, pH 7.4; 100 mM NaCl; 2 mM CaCl_2_ prior to storage at −80 °C. The amino acid sequence of sSORT is provided in Appendix A. The sortilin-binding protein neurotensin, amino acid sequence LYENKPRRPYIL (Genescript, Piscataway, NJ, USA) and neurotensin-Ahx-FITC containing the same sequence with an additional N-terminal modification: FITC-Ahx (Genescript) were used as competitors for binding to sortilin. For performing the assay, fresh 0.1% bovine serum albumin (BSA) was added to the assay buffer to obtain the final concentration: 50 mM HEPES, pH 7.4; 100 mM NaCl; 2 mM CaCl_2_; 0.1% BSA; 0.1% Tween-20. The proteins analyzed in the screen, neurotensin (used as control), IL-6 and IL-8 proteins were serially diluted in ten different concentrations in assay buffer. In each well of a Nunc^®^ MaxiSorp™ 384-well plate (#P6491-1CS, Sigma-Aldrich), 100 nM sSORT were mixed with pre-diluted proteins and 10 nM neurotensin-Ahx-FITC in assay buffer to a final volume of 20 µL. The final concentration range for the proteins included neurotensin (1.5 nM to 30 µM), IL-6 (2.3 nM to 4.5 µM) and IL-8/CXCL8 (5.1 nM to 10 µM). The plate was then briefly centrifuged prior to 1 h incubation at room temperature in the dark. The mPolarization values were obtained from a CLARIOstar plate reader (excitation at 482 nm and emission at 530–540 nm), with each well flashed 200 times. The Z’ value was calculated to 0.79 from a total of 16 positive controls and 16 negative controls.

### 2.8. RNA Sequencing and Data Analysis

This method is described in more detail in [23]. Briefly, total RNA was extracted from MCF7 or MDA-MB-231 cells grown in 2D monolayer or PDS cultures and processed according to the Smart-Seq2 protocol [24], with the addition of an initial hybridization step. Pre-amplification was performed and analyzed on a 2100 Bioanalyzer (Aligent, Santa Clara, CA, USA), and RNA-sequencing libraries were generated using the Nextera XT DNA Sample Preparation and Index kits (both Illumina, San Diego, CA, USA) according to the manufacturer´s recommendations. Single-cell libraries were pooled equimolarly, and libraries were analyzed using 2 × 150 base-paired-end sequencing on a high-throughput sequencer at TATAA Biocenter (NextSeq 500, Illumina). The resulting sequencing reads were aligned to the hg19 reference of the human genome, and gene expression estimates as read counts were obtained and normalized to the sample library size to obtain reads per million (RPM) values.

### 2.9. Patient-Derived Scaffolds (PDSs)

The PDS decellularization and culture system have been described previously [23]. Briefly, decellularized PDSs were sectioned (3 × 3 × 2 mm) and cultured in 48-well plates (Thermo Fisher Scientific) with MCF7 or MDA-MB-231 cells (3 × 10^5^) for 21 days in complete media (DMEM or RPMI, respectively) with 1% Antibiotic-Antimycotic (Gibco^®^). Twenty-four hours after seeding, the PDSs were transferred to new wells with fresh media and then moved every week. At day 16, the PDSs were transferred to new wells with fresh media, and after five days, the media was collected for subsequent analysis. The Regional Research Ethics Committee in Gothenburg approved the processing of patient materials and data (DNR:515-12 and T972-18).

### 2.10. Statistical Analysis

Statistical analysis was performed using GraphPad Prism 8.0 (GraphPad Software), GenEx Software (GenEx 7.0, MultiD Analysis AB) or IBM SPSS Statistics Version 25. Differences between groups were evaluated using two-tailed Student´s *t*-test (comparison between two groups) or one-way ANOVA (comparison of several groups), adjusting for multiple comparisons. Results are presented as the mean ± standard error of mean (SEM). Data were considered statistically significant if *p* < 0.05, where * *p* < 0.05; ** *p* < 0.01; *** *p* < 0.001.

## 3. Results

### 3.1. Progranulin Treatment Promotes Secretion of IL-6 and IL-8 in Breast Cancer

We have previously identified progranulin as a secreted cancer stem cell-propagating factor in breast cancer [12]. Here, we investigated how progranulin treatment affected the secretion of other cytokines, and specifically the secretion mediated via the sortilin receptor, by analyzing conditioned media from MCF7 (estrogen receptor alpha; ERα positive) and MDA-MB-231 (ERα negative) breast cancer cells. Conditioned media were collected 48 h after progranulin treatment in monolayer cultures, with or without the sortilin-binding small molecule AF38469, and secreted target proteins were analyzed using PEA. Overall, analysis of the 184 proteins included in the protein panels revealed distinct secretion profiles for the two cell lines and treatments, where MDA-MB-231 cells showed higher basal secretion compared to MCF7 cells. Differences in secreted proteins in untreated MCF7 cells (vehicle) and after treatment with progranulin and/or AF38469 are illustrated in Figure 1a. Interestingly, IL-6 and IL-8, in parallel with other cytokines and receptors involved in inflammation, including tumor necrosis factor (TNF), chemokine (C-X-C motif) ligand 1 (CXCL1), cluster of differentiation 40 (CD40), Fas ligand (FASL), colony stimulating factor 1 (CSF-1) and tissue plasminogen activator (tPa), were the most upregulated, secreted proteins after progranulin treatment in MCF7 cells (Figure 1b). Additional secreted proteins included in the analyses and the effects of the treatment with progranulin or in combination with AF38469 are presented in Appendix A. Since IL-6 and IL-8 have been associated with cancer stem cell activation in earlier studies, and their secretion was induced by progranulin treatment in both MCF7 (Figure 1) and MDA-MB-231 cell (Appendix A) cultures, we specifically focused on these two cytokines in the further analysis of progranulin-mediated cancer stem cell effects. Inhibition of sortilin using the small molecule AF38469 reduced progranulin-induced IL-6 and IL-8 secretion levels close to the base levels observed in untreated MCF7 cells, suggesting a progranulin-mediated and sortilin-dependent IL-6 and IL-8 secretion (Figure 1c). Similar results were observed for MDA-MB-231 cells and IL-8 secretion, whereas IL-6 secretion was only moderately inhibited, potentially affected by the high basal levels of IL-6 (Appendix A).

### 3.2. A Crosstalk between Progranulin and IL-6/8 Expression

Next, MCF7 and MDA-MB-231 cells were treated with increasing concentrations of progranulin for 48 h, and the endogenous total protein levels of IL-6 and IL-8 were examined using Western blot. Supporting the protein secretion profile in Appendix A, MDA-MB-231 cells showed higher basal protein levels of both IL-6 and IL-8. After progranulin treatment, IL-6 and IL-8 protein levels increased in both cell lines (Figure 2a), validating a progranulin-mediated induction of IL-6 and IL-8 protein levels in breast cancer cells parallel to an increased secretion of the proteins. To determine whether progranulin was influenced by IL-6 and IL-8, cells were treated with increasing concentrations of the cytokines for 48 h, followed by progranulin protein quantification. Interestingly, progranulin protein levels clearly increased after IL-6 treatment (Figure 2b), suggesting a crosstalk between progranulin and IL-6. Similar results were obtained with increasing concentrations of IL-8 (Figure 2c). Taken together, the results revealed an interplay between progranulin and IL-6/8 in breast cancer, demonstrated not only by an increase in secretion of the proteins but also on the cellular protein levels.

### 3.3. IL-6-Mediated Sphere Formation Is Dependent on Sortilin

As both IL-6 and IL-8 have been reported to be involved in the formation and stimulation of the cancer stem cell phenotype in various cancers [3,25], we used the mammosphere formation assay to confirm the effect of IL-6 and IL-8 on the cancer stem cell population in breast cancer. Treatment for 48 h with IL-6 and IL-8 at increasing concentrations led to an increase in the mammosphere-forming capacity of MCF7 cells, with a significant increase in mammospheres with IL-6 at 100 nM (Appendix A). Comparable findings were observed with MDA-MB-231 cells, where both IL-6 an IL-8 induced a significant increase in mammosphere formation at 50 nM (Appendix A). Opposite to the sphere formation capacity, there was no significant difference in cell viability after treatment with increasing concentrations of IL-6 or IL-8 (Appendix A). These results confirm that both IL-6 and IL-8 specifically increase the fraction of breast cancer stem cells, which is in line with previous results.

The progranulin receptor sortilin has recently been identified as a high-affinity receptor for IL-6 [20]. To confirm the binding of IL-6 to sortilin, we performed a competitive binding assay for sortilin using a fluorescence polarization-based method. The results confirmed that IL-6 binds to sortilin by outcompeting the fluorescently labeled neutrotensin, with an IC50 of 11.2 µM (Figure 3a). In contrast to IL-6, IL-8 did not outcompete neurotensin, suggesting that IL-8 does not bind to sortilin, or at least not to the same binding site (Figure 3b). Further, to investigate whether the cancer stem cell expansion triggered by IL-6 and IL-8 was dependent on sortilin, the small sortilin-binding molecule AF38469 was used [26]. MCF7 and MDA-MB-231 cells were treated with IL-6 and IL-8, alone or in combination with AF38469, for 48 h before performing the mammosphere assay. As expected, treatment with IL-6 alone increased the mammosphere capacity of the cell lines. However, when combined with AF38469, there was a reduction in sphere formation for both MCF7 and MDA-MB-231 cells (Figure 3c,e). In contrast, AF38469 did not influence the IL-8-induced mammosphere increase (Figure 3d,f). These results demonstrate that sortilin is required for IL-6- but not IL-8-dependent breast cancer sphere formation in vitro.

### 3.4. Increased Gene Expression of Progranulin, IL-6 and IL-8 in Breast Cancer Cells Adapting to an In Vivo-like 3D Growth System

To further explore and validate the association between IL-6 and the progranulin–sortilin axis in breast cancer, we analyzed the gene expression of associated gene markers in cancer cells grown in an in vivo-like 3D growth system using patient-derived scaffolds (PDS) and compared the results with 2D monocultures [23]. The PDS model has earlier been shown to markedly influence the expression of genes involved in differentiation, epithelial–mesenchymal transition, stemness and proliferation of the adapted cancer cells [23]. Here, we compared alterations in the gene expression of cells grown in monolayer cultures or the PDS model obtained from the dataset described in Landberg et al., where next-generation sequencing was performed [23]. A schematic illustration of the PDS workflow is given in Figure 4a. Transcriptomic analysis of cells showed an upregulation of IL-8 in MCF7 cells, as well as progranulin, IL-6 and IL-8 in MDA-MD-231 cells growing in the human-based scaffold compared to 2D cultures (Figure 4b,c). In addition, there was an increase in sortilin and IL-6 receptor (IL-6R) gene expression, as well as in IL-6 signal transducer (IL-6-ST, also called GP130) in MCF7 cells (Figure 4d). For MDA-MB-231 cells, only IL-6R and IL-6-ST were significantly upregulated (Figure 4e). In terms of the IL-8 receptors, CXCR1 and CXCR2, very low or undetectable levels were observed, with no significant change in the different culture systems (Figure 4d,e). Altogether, the data support the fact that the progranulin–IL-6–sortilin axis was collectively upregulated in cancer cells exposed to human-based growth conditions using the PDS model.

### 3.5. Correlation between IL-6, IL-8 and Progranulin Secretion in Breast Cancer Cells Grown on Patient-Derived Scaffolds

In order to further delineate the relevance of IL-6 and its association with progranulin and sortilin in breast cancer, we analyzed data from Persson et al. in which the PDS model was used to study the secretomic profile of cancer cells grown on patient-specific microenvironments [27]. A detailed secretomic profile from MCF7 and MDA-MB-231 cells grown on 63 individual PDSs, obtained using a high-throughput PEA, was available as illustrated schematically in Figure 5a. In this study, we nevertheless specifically studied the intercorrelation of secreted progranulin, IL-6 and IL-8 with a focus on the potential variability, dependent on the different patients’ microenvironments provided by the individual PDSs. The results clearly showed large variation in the secretion of proteins and cytokines from cancer cells growing in different PDSs (Figure 5b,d). There was further a significant positive correlation between the secretion of IL-6, IL-8 and progranulin in MCF7-PDS cultures (IL-6-progranulin: r = 0.432, *p* < 0.001, IL-6-IL-8: r = 0.336, *p* = 0.011; Figure 5c). In contrast, MDA-MB-231 in the PDS model showed a negative correlation between IL-6 and IL-8 secretion (r = −0.862, *p* < 0.001), as well as between progranulin and IL-8 (r = −0.547, *p* < 0.001; Figure 5e). However, there was a positive correlation between IL-6 and progranulin in MDA-MB-231 cells grown on PDSs (r = 0.312, *p* = 0.023), emphasizing the crosstalk between IL-6 and progranulin (Figure 5e). Scatterplots representing the individual correlations and the variation in secretion are given in Appendix A.

## 4. Discussion

Previous studies have highlighted the importance of the crosstalk between cancer cells and the peritumoral microenvironment during cancer progression, where growth factors, cytokines and other proteins secreted by inflammatory cells, along with other cells in the surrounding stroma, act as messengers between tumor cells and their surroundings [5,6,28]. Cancer cells also produce various cytokines, growth factors and inflammatory receptors that influence the tumor microenvironment, suggesting a possible mechanism for cancer cells to escape an effective immune response, thereby leading to cancer progression [29,30,31]. We have previously described the growth factor progranulin as a cancer cell-secreted factor that induced breast cancer stem cell propagation via the receptor sortilin [12]. Progranulin was secreted at high constitutive levels by the triple-negative breast cancer MDA-MB-231 cells, whereas the ERα-positive MCF7 cells fluctuated in their secretion of progranulin, with an increased secretion under hypoxic conditions [12]. As previously shown, activation of hypoxia-inducible factor 1 (HIF1) in pre-clinical breast cancer models is critical for the pre-metastatic niche formation [32], suggesting that progranulin is indeed involved in forming a favorable niche for cancer stem cell activation and maintenance.

In this study, we have defined a progranulin-induced secretomic profile of breast cancer cells and its contribution to breast cancer stem cell expansion. The altered cell secretion in cancer has earlier been described to influence the formation of the pre-metastatic niche [32,33]. The active systemic secretion from the primary tumor is thought to occur early in cancer progression, even prior to cancer dissemination. At more advanced stages of cancer, the cancer-specific secretome can favor the establishment of metastasis, as well as the survival and proliferation of metastatic cells [34,35]. Here, we evaluated the effect of progranulin treatment on breast cancer cell secretion by examining the presence of various cytokines in the conditioned media of breast cancer cells treated with progranulin, with or without the sortilin-binding molecule AF38469 [26]. In MCF7 cells, progranulin induced secretion of several inflammatory proteins, including cytokines and secreted receptors (Figure 1). Interestingly, several of these progranulin-induced secreted proteins were also suppressed by inhibiting sortilin, suggesting a progranulin-sortilin-mediated secretion (Appendix A). Importantly, many of the progranulin-sortilin-induced secreted proteins have known functions in the formation of metastasis and in the priming of the pre-metastatic niche, such as CXCL1, IL-6 and IL-8 [32,33,34,36]. These findings are in line with our previous discovery that progranulin induced metastasis in vivo via the receptor sortilin [12]. In addition, we could show that progranulin induced secretion of other proteins, independent of sortilin, such as TNF and FASLG. Several of the secreted proteins detected for MCF7 cells were also secreted in MDA-MB-231 cells, such as IL-6, IL-8 and CXCL1 (Appendix A). However, the increase in secretion after progranulin treatment in MDA-MB-231 cells was relatively low compared to MCF7 cells. This could be due to higher basal levels of general secretion, including progranulin, by MDA-MB-231 cells compared to MCF7 cells, thus leaving the cells less responsive to additional exogenous stimuli [37].

Previous studies have shown that IL-6 induced both progranulin gene and protein expression in both cholangiocarcinoma [38] and hepatocellular carcinoma [39]. Progranulin treatment also increases IL-6 mRNA expression in adipose tissue [40]. Nevertheless, no data are available on the potential feedback loop system between these interactions in breast cancer. Here, we demonstrated that treatment with exogenous progranulin induced the protein expression of IL-6 and IL-8 in both MCF7 and MDA-MB-231 cells. By further examining the relationship between IL-6 and progranulin protein expression in breast cancer, we identified a crosstalk between progranulin and IL-6 protein expression, where IL-6 also induced progranulin protein expression (Figure 2). Correspondingly, we observed a similar feedback loop between the protein expression of progranulin and IL-8.

IL-6 has been associated with malignant properties of breast cancer stem cells via the induction of epithelial-to-mesenchymal transition and cell migration and invasion, enabling the establishment of metastasis [41,42]. In addition, IL-6 is important in regulating cancer stem cell self-renewal and the balance between cancer cells and cancer stem cells in several types of cancer [25,41,42,43,44]. Similarly, IL-8 has been demonstrated to affect cancer stem cells in different malignancies [44,45,46,47]. Previous studies have reported an increase in IL-8 serum levels in metastatic breast cancer patients compared to patients with local disease, and elevated IL-8 secretion is also involved in angiogenesis and drug resistance [48,49]. In line with previous reports [48,49,50], we could demonstrate that both IL-6 and IL-8 increased the mammosphere formation of MCF7 and MDA-MB-231 cells, suggesting a more malignant phenotype by an induction of cancer stem cell propagation (Appendix A). By using a competitive sortilin-binding assay, we were able to confirm previous published data that IL-6 binds to the sortilin receptor [20] (Figure 3). However, we did not observe any competition of IL-8 for binding to sortilin, suggesting that IL-8 does not bind to the sortilin receptor, or at least not to the same site, as progranulin and neurotensin. Importantly, and in line with the sortilin-binding capacity, the inhibition of sortilin with AF38469 clearly showed that IL-6, but not IL-8, was dependent on binding to sortilin for mammosphere formation (Figure 3). Taken together, these results suggest that even though secretion of both IL-6 and IL-8 were induced in a progranulin-sortilin-dependent manner, the direct breast cancer stem cell-propagating effect of IL-6 was dependent on sortilin, whereas the IL-8-induced cancer stem cell activation was sortilin-independent. IL-8 probably affects cancer stem cells indirectly via progranulin and IL-6, whereas progranulin and IL-6 can affect cancer stem cells both directly via sortilin and indirectly via associated proteins.

To validate the IL-6, IL-8 and progranulin axis in a more in vivo-like 3D growth system, we utilized the patient-derived scaffold (PDS) model (Figure 4). Compared to conventional 2D monolayer cultures, the 3D tumor microenvironment affects the secretomic and transcriptomic profile of adapting cancer cells, giving rise to an increased cancer stem cell pool [23,51,52]. Further, the patient-derived cancer microenvironments amplified the secretion but also induced a variability of cytokines and growth factors compared to 2D cultures, potentially mimicking actual growth conditions in breast cancer patients. Earlier data also showed that the PDS induction of IL-6 secretion in MCF7 cells correlated with high tumor grade (*p* = 0.004) and patients with lymph node metastasis (*p* = 0.021) [27], supporting the relevance for a link to more aggressive cancer features. 

Here, we also show that IL-6 was positively correlated with progranulin secretion in both MCF7 and MDA-MB-231 cells cultured in the PDS model, further highlighting the possible feedback loop between IL-6 and progranulin (Figure 5). Interestingly, IL-6 was also positively correlated with IL-8 in MCF7 cells, and these cytokines have been linked to the metastatic process, including the priming of the pre-metastatic niche in breast cancer [53,54]. As previously reported, cancer cells adapted to human microenvironments and patient-derived scaffolds became more stem cell-like both regarding induced RNA changes and secretomic profiles [23,27]. The data illustrating that IL-6, IL-8 and the receptors sortilin, IL-6R and IL-6-ST were highly secreted from PDS-adapted breast cancer cell lines support the fact that the PDS model includes a microenvironment promoting the cancer stem cell population. In addition, the PDS model could provide information on patients with an IL-6- and progranulin-inducing microenvironment that could benefit from anti-sortilin therapy [27,55,56,57,58,59].

Exploring the complex network between cancer cells and their tumor microenvironment, including the secretion of various cytokines and growth factors, is of great clinical importance, as it might lead to the identification of new targets and more effective treatment for cancer patients. Current and conventional breast cancer treatments mainly target cancer cells and not the interplay with the surrounding microenvironment, limiting the general effect on disease progression and therapy resistance [7,8]. Monoclonal antibodies targeting IL-6 or relevant receptors have been shown to be effective in the treatment of numerous autoimmune and inflammatory diseases and are now being tested in cancer in combination with other cancer drugs [29]. Therapeutic targeting of IL-6 or progranulin via their common receptor, sortilin, could be a potential novel treatment strategy for cancer forms with high levels of IL-6 or progranulin, inhibiting cancer stem cell functions and consequently reducing tumor aggressiveness and metastasis formation.

## 5. Conclusions

To develop a better understanding of disease progression and response to therapy, we need to reveal the complex cellular communication networks and signaling associated with various cytokines in the tumor microenvironment. Here, we demonstrated a crosstalk between IL-6 and progranulin signaling in breast cancer, involved in the induction of stem cell features through the receptor sortilin. Targeting IL-6 or progranulin or both through sortilin could potentially be a new treatment approach for breast cancer patients with elevated progranulin and/or IL-6 levels, further emphasizing sortilin as an important therapeutic target for treating breast cancer.

## Figures and Tables

**Figure 1 cancers-15-05757-f001:**
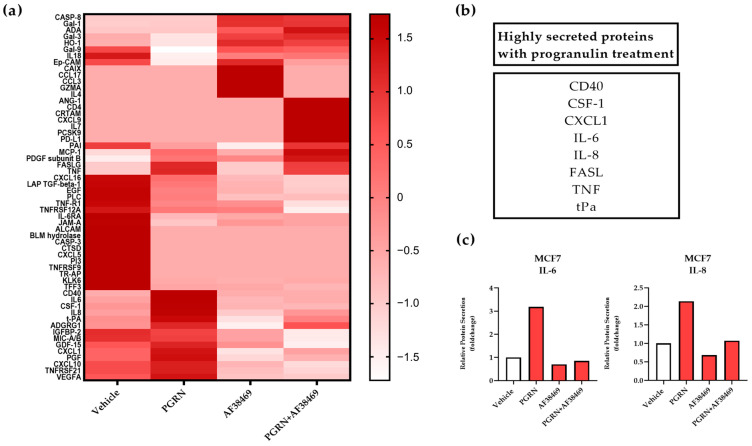
Progranulin treatment increased IL-6 and IL-8 secretion in breast cancer cells. (**a**) Heat map showing secreted proteins and cytokines in MCF7 cells. Cells were untreated (vehicle), treated with 1 μg/mL progranulin or 2 μg/mL AF38469, or a combination, for 48 h. The data are auto-scaled, and each protein is normalized to the total secretion of each sample. (**b**) Table of highly secreted proteins with progranulin treatment. (**c**) Bar charts illustrate the relative fold changes of IL-6 and IL-8 after progranulin treatment in MCF7 cells. Data represent one biological replicate. PGRN: progranulin, IL: interleukin.

**Figure 2 cancers-15-05757-f002:**
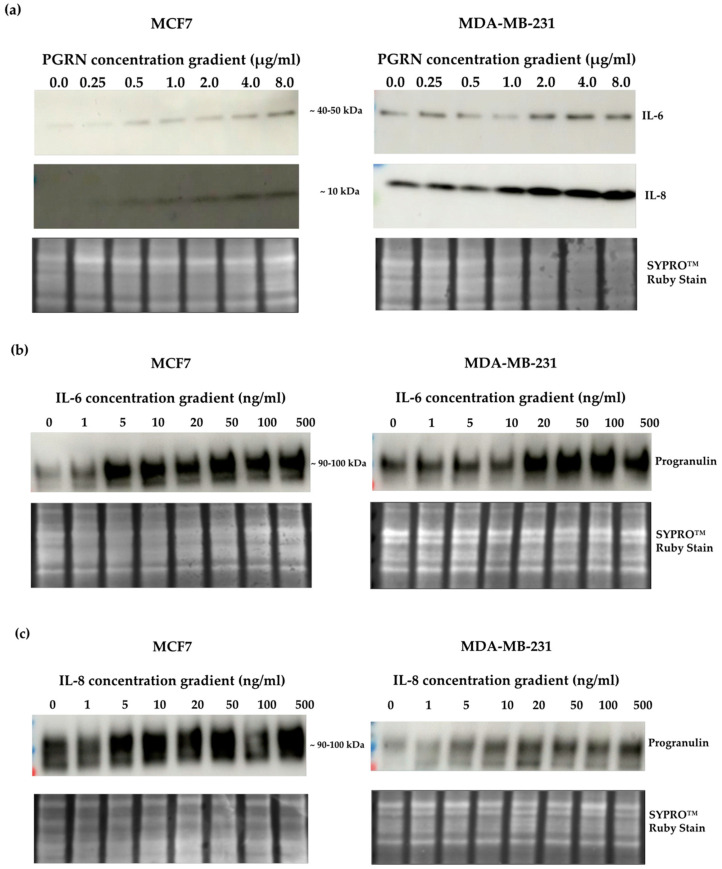
Crosslink between progranulin and IL-6 expression in breast cancer cell lines. Immunoblotting of IL-6, IL-8 and progranulin protein expression in MCF7 and MDA-MB-231 breast cancer cell lines. (**a**) Cells treated with increasing concentrations of progranulin showed an increase in IL-6 and IL-8 expression for 48 h. Treatment with an IL-6 (**b**) or IL-8 (**c**) concentration gradient led to an increased progranulin expression in MCF7 and MDA-MB-231 cells. SYPRO^™^ Ruby (Bio-Rad) stain was used as a loading control. Representative images from one of three independent experiments are shown. Western blot data are cut for visualization purposes, and all samples for the respective antibody were analyzed on the same blot. PGRN: progranulin, IL: interleukin. The uncropped bolts are shown in Appendix A.

**Figure 3 cancers-15-05757-f003:**
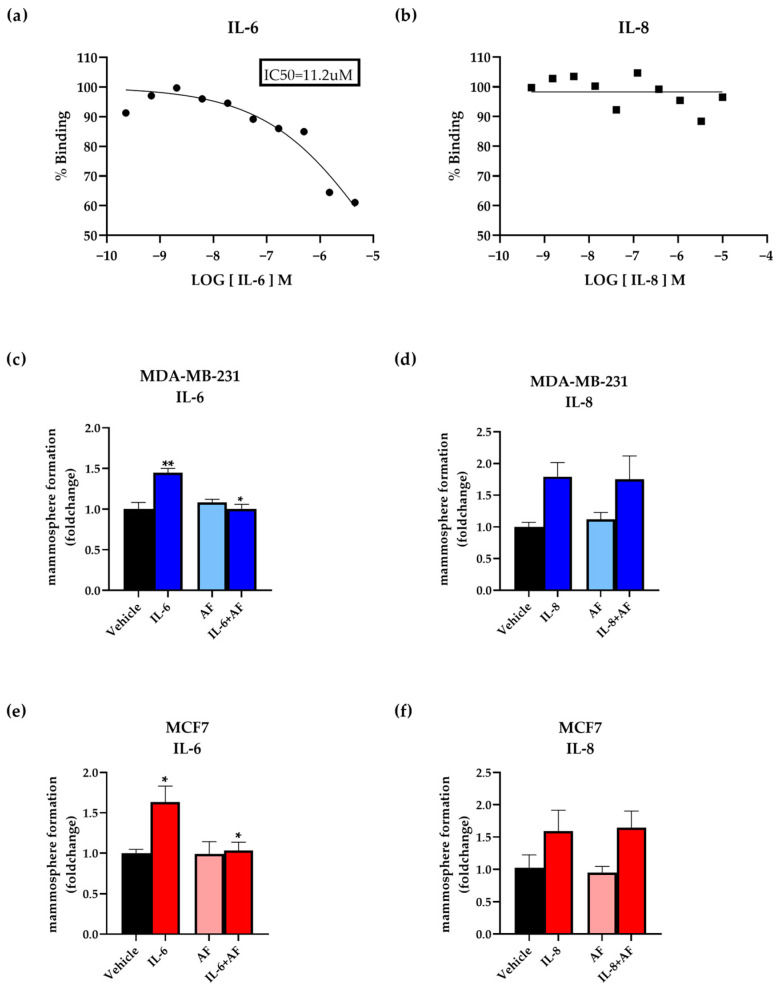
IL-6-induced sphere formation was dependent on sortilin. (**a**,**b**) Fluorescent polarization assay showing that IL-6, but not IL-8, was able to bind to the sSORT receptor and outcompeted neurotensin. Representative image of one run (*n* = 3). Sortilin-dependent mammosphere formation in MDA-MB-231 (**c**,**d**) and MCF7 cells (**e**,**f**). Cells were untreated (vehicle), treated with 100 ng/mL IL-6, 100 ng/mL IL-8 or 2 μg/mL AF38469, or a combination, for 48 h. The bar charts represent the relative number of spheres formed in control samples (vehicle) against various treatments, five days after cell seeding in non-adherent polyHEMA-coated 6-well plates. Results are presented as relative mammosphere formation ± SEM normalized to the control. Statistical significance was calculated using one-way ANOVA adjusted for multiple comparison (*n* = 3), where * *p* < 0.05 and ** *p* < 0.01. PGRN: progranulin, IL: interleukin.

**Figure 4 cancers-15-05757-f004:**
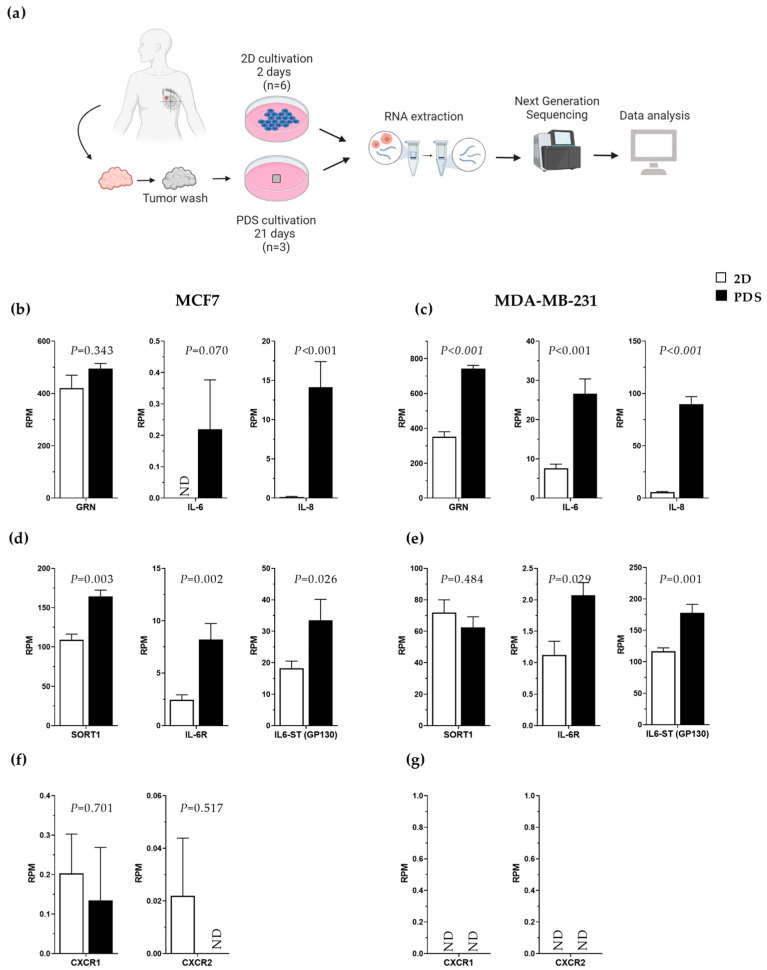
Expression profile of interesting cytokines, proteins and receptors of breast cancer cells grown in 2D monolayer cultures compared to patient-derived scaffolds (PDS). (**a**) Schematic picture showing experimental workflow for next-generation sequencing (NGS) analysis of 2D monolayer cultures (MCF7and MDA-MB-231) compared to PDS cultures (MCF7 and MDA-MB-231). Results indicate transcriptional activation of genes in the PDS (3D) system compared to 2D using MCF7 and MDA-MB-231 cells by performing NGS. Bar plots show an increase in expression of progranulin (PGRN), IL-6 and IL-8 in 2D cultures compared to PDSs in (**b**) MCF7 and (**c**) MDA-MB-231. Differences in gene expression of relevant receptors in 2D cultures compared to PDSs in (**d**,**f**) MCF7 and (**e**,**g**) MDA-MB-231 cells. RPM—reads per million, SORT1 (sortilin receptor—ENSG00000134243.7), GRN (progranulin—ENSG00000030582.12), IL-6 (interleukin-6—ENSG00000136244.7), IL-6R (IL-6 receptor—ENSG00000160712.8), IL-6ST/GP130 (IL-6 signal transducer (also called GP130—glycoprotein 130)—ENSG00000134352.15), IL-8 (interleukin-8—ENSG00000169429.6), CXCR1 (chemokine (C-X-C motif) receptor 1/IL-8 receptor 1—ENSG00000163464.7), CXCR2 (IL-8 receptor 2—ENSG00000180871.3), ND—not detected. Statistical significance was calculated using an unpaired *t*-test (PDSs, *n* = 3 and for 2D, *n* = 6 for both cell lines).

**Figure 5 cancers-15-05757-f005:**
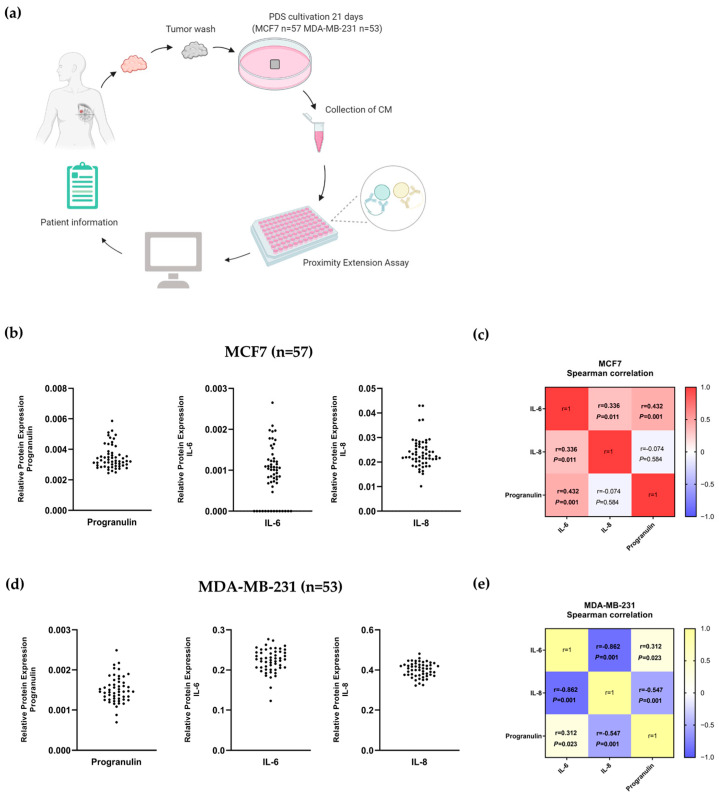
Correlation between secreted IL-6, IL-8 and progranulin in the PDS model system. (**a**) Schematic picture showing experimental workflow for secretome analysis of breast cancer cells grown in PDS. (**b**) Dot plots of the variation in secretion of progranulin, IL-6 and IL-8 for cancer cells grown in the PDS model in MCF7. (**c**) Heat map visualization of pairwise Spearman correlation between different markers in MCF7 cells grown in the PDS culturing system. (**d**) Dot plots of the variation in secretion of progranulin, IL-6 and IL-8 for cancer cells grown in the PDS model for MDA-MB-231. (**e**) Heat map visualization of pairwise Spearman correlation between different markers in MDA-MB-231 cells grown in the PDS culturing system. The data represent MCF7 or MDA-MB-231 cells cultured in PDSs generated from 63 different breast cancer patients (57 for MCF7 and 53 for MDA-MB-231 cells), corresponding to 122 individual samples. *p* < 0.05 is considered statistically significant. PDS: patient-derived scaffold, r: correlation coefficient, IL-6: interleukin-6, IL-8: interleukin-8, CM: conditioned media.

## Data Availability

The data generated and analyzed during the current study are available from the corresponding author upon reasonable request.

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
