# Peer review of "Interleukin-6 Induces Stem Cell Propagation through Liaison with the Sortilin–Progranulin Axis in Breast Cancer"

_cancers, 2023, doi:10.3390/cancers15245757_

Round 1

Reviewer 1 Report

Comments and Suggestions for Authors

This study reported a potential functional correlation between the growth factor progranulin and cytokines interleukin- 6 and -8 that promote breast cancer stemness, which also may involve one of the receptors of progranulin, sortilin. Overall, the quality and amount of the data failed to convince the reviewer and did not support some of the conclusions they claimed. Functional studies were focused on previous knowledge and did not make new conclusions (Figure 3) while the mechanism of the induction of cytokines was unexplored. Two breast cancer cell lines MCF-7 and MDA-MB-231 were majorly used in this study, however, some data between these two cell lines were inconsistent (Figure 2a, Figure 3b and 3d, Figure 5b-e), suggesting a validation in a third cell line is needed. In addition, another major weakness of this study is too many correlation analyses have been used whereas lack of proper function validations. A large amount of new experimental evidence and validation studies are needed to support their conclusions. 

Figure 2a does not suggest IL-6 was induced by progranulin in breast cancer, the induction was minor and did not show a dose-dependent manner. In addition, the IL-6 induction by 1 μg/ml progranulin from the western blot shown in Figure 2a is not consistent with the heatmap shown in Figure S2. A repeat experiment for IL-6 induction by progranulin treatment in MDA-MB-231 is required. Otherwise, a third cell line is recommended for these assays.

Comments on the Quality of English Language

Proper English language was used in this manuscript and I have no further comments.

Author Response

1) Functional studies were focused on previous knowledge and did not make new conclusions (Figure 3) while the mechanism of the induction of cytokines was unexplored.

We agree with the reviewer that the effect of the cytokines IL-6 and IL-8 on the cancer stem cell population has been described before and is accordingly acknowledged in the discussion section of the manuscript (lines 487-496). With this mammosphere formation functional assay, we aimed at verifying the effect of both cytokines in our breast cancer cell models used. However, we would like to mention that to our knowledge, the following functional study showing that the effect of IL-6 on the cancer stem cell population is through the sortilin receptor (Figure 4) is novel and we believe is of importance for anti-cancer drug development, especially for aggressive cancers with high levels of progranulin and/or IL-6.

Action taken: Figure 3 has been moved to supplementary (now named Figure S3) since as mentioned by the reviewer it does not show significant data but rather supports our results as well as previous ones described by others. Original sections 3.3 and 3.4 have been merged into one section.  

2) Two breast cancer cell lines MCF-7 and MDA-MB-231 were majorly used in this study, however, some data between these two cell lines were inconsistent (Figure 2a, Figure 3b and 3d, Figure 5b-e), suggesting a validation in a third cell line is needed.

We believe the reviewer refers to the differences found between the two cell lines studied and not differences within the same cell line in different experiments. We agree with the reviewer that these two cell lines showed different basal levels of intracellular cytokines and growth factors, which led to different response thresholds to exogenous progranulin. More specifically, MCF7 cells showed lower endogenous levels of progranulin, IL-6 and IL-8, which made them more sensitive to low concentrations of exogenous protein exposures. Furthermore, we never expected to obtain the exact same results using two quite different breast cancer cell lines regarding genetic changes, subtype origin and proliferative and infiltrative properties. Despite this, both breast cancer cell lines responded similar, but MDA-MB-231 cells needed higher doses to trigger changes. Since MCF7 cells possess lower intracellular levels of endogenous progranulin, IL-6 and IL-8 proteins, the effect of blocking sortilin using this cell line produced more moderated and coherent data and was therefore considered to be the favorable model system producing illustrative and representative data for this study.  

Action taken:

The sentence in line 253 (results section) has been rewritten as “Similar results were observed for MDA-MB-231 cells and IL-8 secretion whereas IL-6 secretion was only moderately inhibited potentially affected by the high basal levels of IL-6 (supplementary Figure 2).”

The sentence in line 473 (discussion section) has been rewritten to clarify the lower cytokine secretion after progranulin treatment in MDA-MB-231 cells compared to MCF7 cells. The sentence is as follows “This could be due to higher basal levels of general secretion, including progranulin, by MDA-MB-231 cells compared to MCF7 cells, thus leaving the cells less responsive to additional exogenous stimuli.”

With the above changes and clarifications, we are convinced that two cell lines are sufficient to produce the first proof of concept data within the study then further backed up and validated with the sections including a large cohort of patient derived scaffolds (Figures 4 and 5).

3) Another major weakness of this study is too many correlation analyses have been used whereas lack of proper function validations. A large amount of new experimental evidence and validation studies are needed to support their conclusions.

We appreciate the reviewer’s observation since it has been very useful to see that the information could be easily misinterpreted. We used the patient-derived scaffold (PDS) system as an in vivo-like 3D-growth platform to validate the results. PDSs have the intact extracellular matrix (ECM) and associated molecules, keeping relevant and unique information from the tumor microenvironment of each patient. Published data demonstrated that PDSs can be repopulated with standardized cancer cell lines, and the unique microenvironment induced changes in the adapting cells in a more clinically relevant manner (PMID: 33356003, 31978840, 34172801, 33368325). Additionally, the correlations shown in section 3.6 are from secreted IL-6, IL-8, and progranulin from MCF7 and MDA-MB-231 cells grown in PDSs, supporting the above-mentioned data.

Action taken: 

Sentence in line 363 (results section) has been rewritten to indicate the use of the PDSs to validate the results found. The sentence is as follows “To further explore and validate the association between IL-6 and the progranulin-sortilin axis in breast cancer, we analyzed the gene expression of associated gene markers in cancer cells grown in an in vivo-like 3D growth system using patient-derived scaffolds (PDS) and compared the results with 2D monocultures”

Sentence in line 513 (discussion section) has been rewritten to specify the use of PDSs to validate previously presented data, as follows “To validate the IL-6, IL-8 and progranulin axis in a more in vivo-like 3D growth system, we utilized the patient-derived scaffold (PDS) model (Figure 4).”.

4) Figure 2a does not suggest IL-6 was induced by progranulin in breast cancer, the induction was minor and did not show a dose-dependent manner. In addition, the IL-6 induction by 1 µg/ml progranulin from the western blot shown in Figure 2a is not consistent with the heatmap shown in Figure S2. A repeat experiment for IL-6 induction by progranulin treatment in MDA-MB-231 is required. Otherwise, a third cell line is recommended for these assays.

We thank the reviewer for her/his observation, and we apologize for the incautious general dose-dependent effect statement. While we agree that IL-6 was not induced in a dose-dependent manner in MDA-MB-231 cells (for the reason previously stated that their threshold response is higher due to endogenous progranulin), MCF7 cells showed an increased IL-6 expression more similar to a dose-dependent manner after progranulin treatment. We are aware that western blot is a semi-quantitative technique that provides a relative comparison of protein levels among the lanes, thus we will not qualify the increase as “dose-dependent”. In relation to that, we believe that a quantitative comparison between intracellular proteins obtained by means of western blot (Figure 2a) and secreted proteins to the media showed in the heatmap (Figure S2) is very adventurous. Both results rather than inconsistent, are supporting each other. After progranulin treatment, the intracellular expression (western blot in Figure 2a) and secreted IL-6 and IL-8 (heatmap Figure S2) are increased in MDA-MB-231 cells compared to untreated or vehicle (please notice that the order of treatments in both heatmaps is different).

Action taken: the “dose-dependent” statement has been removed from the manuscript (lines 273, 280) and in Figure 2 footnote (line 291).

Reviewer 2 Report

Comments and Suggestions for Authors

The study entitled “Interleukin-6 Induces Stem Cell Propagation Through Liaison with the Sortilin-Progranulin Axis in Breast Cancer” demonstrates crosstalk between IL-6 and progranulin signaling in the induction of stem cell features through the receptor sortilin in breast cancer. The study is well designed; however, there are a few major concerns that need to be addressed before it could be accepted for publication.

Major comments:

1.     In addition to figures 2b and c, authors need to show combinatorial effect of IL-6/8 and AF38469 (sortilin inhibitor) on both MCF-7 and MDA-MB-231 to validate the feedback mechanism of IL-6/8 on progranulin protein expression.

2.     In figure 3, the validation for increase in mammosphere formation also needs to be validated in combination treatment of IL-6/8 and AF38469 (sortilin inhibitor) on both MCF-7 and MDA-MB-231 cells.

3.     In Supplementary Figure 3a-d, the difference in cell viability with increasing concentrations of IL-6 or IL-8 needs to be validated with an additional cell viability assay/method such as clonogenic assay.

Comments on the Quality of English Language

 Extensive editing of English language required

Author Response

1) In addition to figures 2b and c, authors need to show combinatorial effect of IL-6/8 and AF38469 (sortilin inhibitor) on both MCF-7 and MDA-MB-231 to validate the feedback mechanism of IL-6/8 on progranulin protein expression.

We appreciate the reviewer’s suggestion. We believe that with the results shown in Figure 2 we can already state that there is a feedback mechanism between IL-6/IL-8 on progranulin expression. Using the AF38469 sortilin inhibitor as mentioned by the reviewer would give us an idea of the role of sortilin receptor in this feedback. For that and knowing the importance of IL-6 and IL-8 cytokines in the mammosphere formation capacity of these cell lines (also shown in the newly included Figure 3 to the supplementary material), we performed the functional mammosphere assay including AF38469 in the following section (current 3.3 section). We believe the reviewer’s concern depend on the fact that this is an indirect assay showing the role of sortilin in IL-6 and IL-8 expression. For that reason, as shown in paragraph starting in line 334, we performed a direct assay. That is, a competitive binding assay for sortilin using a fluorescence polarization-based model (current Figure 3a and b). Using this assay we can conclude as mentioned in this 3.3 section that even though there is a feedback mechanism between IL-6/IL-8 with progranulin, only IL-6 is dependent on the sortilin receptor. Altogether, we showed the importance of progranulin, IL-6 and sortilin in the cancer stem cell population, cells that play a central role in tumor growth, heterogeneity, resistance to treatment, and in the ability of the tumor to recur and metastasize, creating a rational to develop future novel strategies targeting sortilin receptor for both IL-6 and progranulin expressing tumors.

2) In figure 3, the validation for increase in mammosphere formation also needs to be validated in combination treatment of IL-6/8 and AF38469 (sortilin inhibitor) on both MCF-7 and MDA-MB-231 cells.

We thank the reviewer for her/his comment. We would like to refer back to Figure 4c-f (now named Figure 3) where we show the effect on mammosphere formation for the combination of IL-6 and IL-8 with the sortilin inhibitor AF38469. Briefly, AF38469 treatment reduced the mammosphere-formation effect caused by IL-6 treatment but not by IL-8, therefore we state in line 347 (result section) that “these results demonstrated that sortilin in required for IL-6, but not IL-8-dependent breast cancer sphere formation in vitro”.

Action taken: in order to emphasize the results mentioned by the reviewer, we have moved Figure 3 to supplementary material thereby focusing to the results in the former Figure 4.

3) In Supplementary Figure 3a-d, the differences in cell viability with increasing concentrations of IL-6 or IL-8 needs to be validated with additional cell viability assay/method such as clonogenic assay.

We appreciate the reviewer’s suggestion.

Action taken: we have plotted the number of cells counted after treatment used prior mammosphere formation assay (see section 2.5). Briefly, cells are seeded in adherent plates with complete media and left for 24 hours to allow attachment. After that, cells were treated for 48 hours, trypsinized, counted and cultured in non-adherent plates with serum-free media. This additional cell viability indicator has been included to the former Figure S3 showing the alamarBlue viability assay, now named Figure S4. Additionally, and as suggested by the reviewer we have performed the clonogenic assay using the maximum cytokine (IL-6 or IL-8) dose used in the mammosphere formation assay (Revision Figure 1). We have included in the assay a third breast cancer cell line (T47D cells). Cells have been seeded and treated as performed for the mammosphere formation assay, therefore the number of cells have also been represented (Revision Figure 1A), showing similar results as those included in the new Figure S4 from the manuscript. Revision Figure 1B and 1C show that MDA-MB-231, MCF7 and T47D breast cancer cells treated with 500 ng/mL IL-6 or IL-8 formed similar number of colonies than the control (0 ng/mL), indicating similar proliferation, survival and reproductive integrity of treated single cells.

Revision Figure 1. Effect of IL-6 and IL-8 on colony formation in breast cancer cells. A) Graph bar indicating the number of MDA-MB-231, MCF7 and T47D breast cancer cells after treatment with 500 ng/mL of IL-6 or IL-8 relative to untreated (0 ng/mL). Cells were counted using a MOXI Z automated cell counter. B) Graph bar showing the number of total colonies formed by MDA-MB-231, MCF7 and T47D breast cancer cells after treatment with 500 ng/mL of IL-6 or IL-8 and untreated (0 ng/mL). Colonies were fixed with 4% paraformaldehyde and stained with crystal violet. C) Representative images from the colonies formed by MDA-MB-231, MCF7 and T47D breast cancer cells after treatment with 500 ng/mL of IL-6 or IL-8 and untreated (0 ng/mL).

Reviewer 3 Report

Comments and Suggestions for Authors

The manuscript by Berger et al. demonstrates that progranulin, a breast cancer secreted protein, increases the production/secretion of IL-6 and IL-8 in both ER-positive and ER-negative breast cancer cell lines.  The augmented secretion of the two cytokines increases the number of cancer stem cells.  Overall, the study is well conceived, structured and presented.  The present study does not require further experimental work.   

Comments on the Quality of English Language

The english language requires minor revisions by an english mother-language editor.

Author Response

We appreciate the time and effort dedicated and the positive feedback on the manuscript.

Round 2

Reviewer 1 Report

Comments and Suggestions for Authors

The authors have sort of addressed my concerns, although no additional experimental validations were provided as suggested, they gave considerable explanations. I agree to publish the current form of the manuscript.

Reviewer 2 Report

Comments and Suggestions for Authors

Authors have addressed all the comments. The revised manuscript could be considered for the publication in cancres. 

Comments on the Quality of English Language

Minor editing of English language required